# Modulation of NK Cell Properties by ESKAPE Group Bacteria

**DOI:** 10.3390/ijms26178449

**Published:** 2025-08-30

**Authors:** Polina Grebenkina, Varvara Juchina, Elizaveta Tyshchuk, Ananstasia Gulina, Elizaveta Denisova, Zoia Korobova, Sergey Orlov, Areg Totolian, Lyudmila Kraeva, Dmitry Sokolov

**Affiliations:** 1Saint-Petersburg Pasteur Institute, St. Petersburg 197101, Russia; grebenkinap@gmail.com (P.G.);; 2Research Institute of Obstetrics and Gynecology named after D.O. Ott, St. Petersburg 199034, Russia; 3Department of Immunology, Pavlov First State Medical University of St. Petersburg, L’va Tolstogo St. 6-8, St. Petersburg 197022, Russia; 4Federal State Budgetary Scientific Institution ‘Institute of Experimental Medicine’, St. Petersburg 197022, Russia

**Keywords:** cancer, cytotoxicity, ESKAPE group, maternal-fetal interface, natural killer (NK) cells

## Abstract

Functions of natural killer cells (NK cells) can be modulated by environmental stimuli. However, the role of bacterial components in this modulation remains an area of ongoing research. This study investigates how bacterial supernatants influence NK cell function—including cytotoxicity, phenotype, and cytokine and mRNA production—following co-culture. These parameters were measured by flow cytometry and RT-PCR. We found that NK cells express TLR2 and TLR5, and that exposure to supernatants from ESKAPE group bacteria modified their cytotoxicity against JEG-3 and K-562 cell lines and their *NKG2A* and *IL-10* mRNA levels. Our findings indicate that bacteria can modify NK cell features and their interactions with other cells in the microenvironment.

## 1. Introduction

Natural killer cells (NK cells) are a type of lymphocyte that play an important role in innate immunity. They exhibit cytotoxic activity against malignant cells and contribute to antitumor immunity. A resident population of NK cells, known as decidual NK cells, exists in certain tissues such as the uterus. These cells form the majority of uterine lymphocytes and play a crucial role in physiological pregnancy by interacting with trophoblast cells and establishing immunological balance at the mother–fetus interface. The ESKAPE group of bacteria includes six bacterial species: *Enterococcus faecium*, *Staphylococcus aureus*, *Klebsiella pneumoniae*, *Acinetobacter baumannii*, *Pseudomonas aeruginosa*, and *Enterobacter species*. These bacteria are known for their ability to develop multidrug resistance and are responsible for a significant number of nosocomial [1] and other infections: bacteremia [2,3,4], endocarditis [5,6], toxic shock syndrome [7], respiratory system diseases [8,9,10,11], urinary, and genital tract infections [12,13,14,15].

The ESKAPE bacterial group may contribute to the development of various pathological conditions related to changes in NK cell functional activity. These bacteria can also be found in the tumor microenvironment [16,17], which provides an opportunity to study the pathogenesis of cancer from a different perspective and discover new therapeutic targets.

NK cells are present in the tumor microenvironment as well, but their cytotoxic activity is reduced [17]. It is currently believed that both other immune cells in the tumor environment and the tumor cells themselves may contribute to this reduction [18]. Tumor cells, for instance, can directly inhibit NK cell function by expressing stress molecules such as MICA/B on their surface. These molecules are ligands for NKG2D on NK cells, and their binding triggers cytotoxic activity. However, tumors can produce soluble forms of the MICA/B molecules, which also bind to the NKG2D receptor. Instead of triggering tumor cell lysis, this reduces the cytotoxic activity of NK cells [19]. In addition, bacterial involvement in this process is also possible. Bacterial metabolites can create an immunosuppressive environment that promotes tumor growth, as reported in studies by other researchers [17,20]. The elimination of intracellular bacteria has been shown to enhance the efficacy of immunotherapy for malignant tumors by counteracting the effect of the immunosuppressive microenvironment on resident immune cells [21].

In addition, microorganisms have been identified in various reproductive dysfunction disorders. Dysbiotic alterations in the microbiome of the reproductive system in pregnant women have been considered a factor that enhances the risk of pregnancy termination [22,23]. Treatment of these conditions is complicated by the presence of antibiotic-resistant bacteria from the ESKAPE group [24]. Infections caused by *S. aureus* and *E. faecalis* can provoke aerobic vaginitis development, which contributes to inflammatory processes during pregnancy and adversely affects the intrauterine fetal development [25]. Research indicates that in women with reproductive problems, multi-drug resistant strains of *E. faecalis* and *P. aeruginosa* are frequently found in the endometrial tissue [26]. Increased levels of NK cells in peripheral blood, in combination with colonization of the vagina by Gram-negative anaerobic bacteria, including *Enterobacter and Klebsiella* spp., have been associated with recurrent cases of spontaneous pregnancy loss [27]. Based on the results of a retrospective study conducted in 2019, infectious conditions of the genital tract, such as endometritis caused by ESKAPE group bacteria (*Enterobacter* spp., *K. pneumoniae*, *S. aureus*, *P*. *aeruginosa*), have been linked to reproductive disorders. Specifically, 74% of women who experienced these infections had a history of early pregnancy loss [14]. Additionally, women with a history of recurrent miscarriages tend to have an endometrial microbiota dominated by *Acinetobacter* species [28]. Some reports show that other ESKAPE bacteria (*S. aureus*, *K. pneumoniae*) can also be linked with pregnancy loss [29,30].

Based on the above evidence, ESKAPE group bacteria have the potential to alter the properties of NK cells, interfering with their interactions with other cells in the host and contributing to the development and progression of pathological conditions. Therefore, investigating the effects of prokaryotic organisms on NK cells is essential—not only to advance our understanding of the molecular mechanisms driving disease pathogenesis but also to develop new therapeutic strategies aimed at modulating NK cell function.

## 2. Results

### 2.1. Effects of Bacterial-Derived Supernatants on the Development of Apoptosis Stages in NK-92 Cells

After culturing NK-92 cells for 24 h in the presence of bacterial supernatants, there was no decrease in the number of viable cells and cells at the stage of late apoptosis or necrosis (Figure 1A,C,D). However, cultivation in the presence of supernatants from K. pneumoniae, Enterobacter spp., or S. aureus led to an increase in the number of NK-92 cells stained only with YoPro, i.e., in a state of early apoptosis (Figure 1B).

### 2.2. Changes in the Cytotoxic Activity of NK-92 Cells After Supernatant Exposure

Pre-incubation of NK-92 cells with *K. pneumoniae*, *Enterobacter* spp., and *S. aureus* supernatants decreased their cytotoxic potential, resulting in a decrease in the number of dead cells of the K-562 line relative to cultivation with intact effector cells (Figure 2).

Pre-incubation of effector cells with *E. faecium*, *A. baumannii*, and *Enterobacter* spp. supernatants led to an increase in the cytotoxic potential of NK-92 cells, which resulted in an increase in the number of dead JEG-3 cells relative to cultivation with intact effector cells (Figure 3).

### 2.3. Changes in the Cytotoxic Activity of NK Cells in PBMC with Supernatants

Based on the results described in Section 2.2, for further experiments, supernatants from three bacterial species were used: *K. pneumoniae*, *A. baumannii*, and *Enterobacter* spp. Their distinct effects determine the choice of these supernatants. The *A. baumanii* supernatant did not affect the cytotoxicity of NK cells against K-562 but increased when JEG-3 cells were used as targets. *K. pneumoniae* supernatant, on the contrary, reduced cytotoxicity against JEG-3 without affecting the effector functions of NK cells against K-562 cells. The *Enterobacter* spp. supernatant led to a change in the cytotoxicity of NK cells in relation to both K-562 and JEG-3 cells, but in different directions of reducing and enhancing, respectively.

Pre-cultivation of PBMC in the presence of *Enterobacter* spp. supernatants led to a decrease in the cytotoxic activity of NK cells in PBMC (Figure 4A) against cells of the K-562 lineage (at the effector ratio (PBMC):target 10:1). In addition, NK cells constitute approximately 10% of the PBMC fraction, so we decided to increase the amount of PBMC in the experiments to increase the effector ratio (PBMC): the target is up to 20:1. In this variant, a decrease in the number of dead cells of the K-562 line was also noted when cultured with PBMC previously incubated in the presence of *A. baumannii* and *Enterobacter* spp. supernatants (Figure 4B).

It was found that NK cells in PBMC altered cytotoxic activity against JEG-3 cells after preliminary incubation in the presence of supernatants. The number of dead JEG-3 cells turned out to be higher after cultivation with PBMC, previously incubated with *A. baumannii, Enterobacter* spp., *K. pneumoniae* supernatants, relative to intact PBMC (Figure 5).

### 2.4. Bacterial Supernatant-Induced Changes in Cytokine Gene mRNA Profiles in NK-92 Cells

*K. pneumoniae* supernatant resulted in a decrease in the relative mRNA content of the gene encoding *IL-10* but did not lead to a change in the relative mRNA content of *IFNy* or *RANTES* (Figure 6).

### 2.5. Unchanged Cytokine Profile in NK-92-Conditioned Media Following Bacterial Supernatant Treatment

Cultivation of NK-92 cells in the presence of bacterial supernatants did not affect the cytokine content in conditioned NK cell media. Cultivation of NK-92 cells in the presence of the stress-inducing factor PMA led to an increase in the content of IL-10, IFNy, and RANTES in conditioned media (Figure 7). In addition, culturing NK-92 cells in the presence of the pro-inflammatory cytokine TNFα led to an increase in the content of IFNγ and RANTES in conditioned media.

### 2.6. Bacterial Supernatants Alter the Relative mRNA Content of Receptor-Encoding Genes in NK-92 Cells

Pre-cultivation of NK-92 cells in the presence of *A. baumannii, Enterobacter* spp., and *K. pneumoniae* supernatants did not lead to a change in the relative number of NKG2C and NKp30 receptor mRNAs (Figure 8). Cultivation of NK-92 cells in the presence of *K. pneumoniae* supernatant resulted in a decrease in the relative content of NKG2A mRNA.

### 2.7. NK-92 Cells Express TLR Family Molecules

Flow cytometry revealed that NK-92 cells spontaneously express TLR2 and TLR5, but not TLR1, TLR4, or TLR6 (Figure 9 and Figure 10). Expression was induced only upon PMA stimulation for TLR5, an effect observed in both the percentage of positive cells and the MFI (Figure 9 and Figure 10). Neither cytokine, LPS, nor bacterial-derived supernatants had an effect on the expression of other TLRs.

### 2.8. NK-92 Cells Contain mRNAs of TLR2 and TLR5 Molecules, as Well as TLR1 and TLR6

The relative number of TLR family surface receptor mRNAs in intact NK-92 cells was estimated. It was found that NK-92 cells contain TLR1, TLR2, TLR5, and TLR6 receptor mRNAs (Figure 11, Appendix A).

## 3. Discussion

Microenvironmental factors can affect the properties of immune cells, but the specific effects of bacteria on NK cells have not been studied extensively. The initial stage of this work involved analyzing the effects of bacteria from the ESKAPE group on NK cells.

To evaluate NK cell responses, optimal concentrations of bacterial supernatants were established by assessing NK-92 cell viability through flow cytometry and propidium iodide staining. Staining cells with this dye indicates their transition to a necrotic or late apoptotic state [31], as it can penetrate the cell membrane if it is compromised [32], and is a commonly used method for assessing cell death [33,34]. To better evaluate supernatant effects on cell viability without inducing cytotoxicity, we tested diluted supernatants and assessed viability using YoPro/PI dual staining. This allows us to identify cells that have undergone early and late apoptosis [35]. Specific staining patterns distinguish the different stages of cell death. For example, YoPro can penetrate a cell immediately after the disruption of membrane integrity [36], while PI can only penetrate a cell at later stages of death.

Supernatant from *S. aureus*, *K. pneumoniae*, and *Enterobacter* spp. can cause an increase in cells at the early apoptosis stage but do not affect the overall number of viable NK-92 cells. The induction of apoptosis in eukaryotic cells is a common mechanism of bacterial pathogenicity [37,38]. It has been previously shown that *P. aeruginosa* can initiate apoptosis in NK cells through both caspase-9-dependent and mitochondrial pathways after intracellular penetration [39]. However, it is worth noting that early apoptosis is a reversible process. When DNA damage has been repaired and the stressor has been removed, the cell can stop the process of self-destruction by restoring its structural integrity [40,41,42]. The results obtained from YoPro staining confirm the selection of supernatant concentrations for further experiments.

Despite the increase in early apoptosis, NK-92 cells exposed to supernatants retained cytotoxic function against both K-562 and JEG-3 cells. We used these two lines to model interactions with tumor cells and fetal trophoblast cells, respectively.

K-562 cells exhibit characteristics similar to those of chronic leukemia cells [43] and serve as a model for studying the interaction between immune system cells and tumors [44,45]. A significant aspect of the current research on antitumor immunity involves the influence of bacteria on tumor development. In this study, we analyzed the cytotoxic effects of NK-92 cell-induced *K. pneumoniae*, *Enterobacter* spp., and *S*. *aureus* supernatants on target cells derived from the K-562 line. A decrease in contact cytolytic activity compared to that of intact effector cells was observed A similar inhibitory effect was observed for PBMC at a 20:1 effector-to-target ratio. The supernatants from *A. baumannii*, *K. pneumonia,* and *Enterobacter* spp. exhibited cytotoxicity-inhibitory effects, which may represent mechanisms of tumor growth in the presence of these bacteria in vivo. ESKAPE bacteria are associated with increased mortality among cancer patients [6]. It has been shown that bacterial infections of the oral cavity, particularly those associated with *Fusobacteria*, can contribute to tumor development. In the presence of polymicrobial communities, there is an increase in matrix metalloproteinases, which may indicate increased invasive potential and a rise in the synthesis of IL-8, a proangiogenic factor that promotes tumor growth [46]. Furthermore, some bacterial antigens have been shown to suppress the effector functions of lymphocytes. For example, *Fusobacterium nucleatum*, which has been linked to colorectal and breast cancer, interacts with TIGIT and CEACAM1 on the surface of CD4+ T cells, inhibiting their cytotoxic potential. Additionally, exotoxin A produced by *P. aeruginosa* reduces the cytotoxic activity of NK cells in PBMC. Another study found that when NK cells in PBMC are cultured in the presence of *P. aeruginosa*, they reduce the expression of CD107a, an indirect indicator of cytotoxic capacity.

When analyzing changes in the cytotoxic activity of NK cells in PBMC fraction, two culturing options were used: a ratio of effector to target cells of 10:1 and 20:1. According to the literature, both options are considered optimal [47]. However, our findings suggest that using different cell numbers leads to different outcomes.

With a slight advantage for the effector cells (ratio of 10:1), the cytotoxic activity of NK cells increased in the presence of Enterobacter species. Conversely, increasing the number of NK cells relative to the target cells to a 20:1 ratio resulted in a decrease in cytotoxic potential when exposed to supernatants.

In addition to NK cells, PBMC also contains other cell types, such as monocytes, T lymphocytes, and dendritic cells. Intercellular interactions within this complex microenvironment may also be affected by bacterial supernatants, potentially influencing the response of NK cells. Since bacteria can inhibit the immune response not only of NK cells but also of other cells, it is possible that a larger number of mononuclear cells would be more effective in this process, leading to a reduction in the cytotoxic potential of the fraction. The activation of T-regulatory lymphocytes has been demonstrated, which, in response to bacterial presence, leads to the deactivation of NK cell cytotoxicity [48]. Conversely, a smaller number of PBMC does not allow bacterial factors to fully exert their suppressive effect on PBMC. This phenomenon warrants further investigation.

For subsequent experiments on trophoblast cell interaction, the number of NK-92 cells remained unchanged. However, when working with PBMCs, an effector-to-target ratio of 20:1 was chosen.

When analyzing the cytotoxic activity of NK-92 cells and PBMC against trophoblast cells of the JEG-3 line, it was noted that the efficiency of contact cytolysis induced by *E. faecium*, *A. baumannii*, and *Enterobacter species* has increased. Dysbiosis of the maternal reproductive system has been shown to be a factor that increases the risk of pregnancy loss [22,23]. *P. aeruginosa* has been identified as one of the causes of miscarriage and sepsis in pregnant women and newborns [14,49]. It has also been found that multidrug-resistant *Enterococcus faecalis* and *Pseudomonas aeruginosa* can be present in the endometrium of women with reproductive disorders [26]. It is possible that these bacteria of the ESKAPE group disrupt the balance required for optimal interaction between trophoblast and NK cells, potentially leading to impaired fertility or pregnancy complications.

The cytotoxic activity of NK cells varies depending on the target cell type. This is likely due to the specific receptor repertoire of the effector cells, which can lead to a range of outcomes, including a cytotoxic response. K-562 cells have been shown to have low expression of ligands on their surface that can inhibit NK cell activity [50], whereas JEG-3 cells have both inhibitory and activating ligands for NK cells. These differences in ligand expression may contribute to the different responses seen in these two cell types when exposed to NK cells [51,52,53].

Based on the results shown in Section 2.2, in order to further investigate the effect of bacterial supernatants on NK cell characteristics, three species were utilized: *K. pneumoniae*, *A. baumannii*, and *Enterobacter* spp. These supernatants were chosen based on their observed effects: the *A. baumanni* supernatant had no effect on NK cell cytotoxicity against K-562 targets but enhanced it when JEG-3 cells were utilized. In contrast, the *K. pneumoniae* supernatant reduced JEG-3 cytotoxicity without altering NK cell effector functions against K-562. The *Enterobacter* supernatant caused changes in NK cell cytotoxicity towards both K-562 and JEG-3, but in opposite directions: reduction and enhancement, respectively. These differences in effects on NK cytotoxicity have not previously been described. Additionally, the impact of three supernatants on a complex of NK cell features was assessed: alterations in the phenotype, mRNA content, and secretory pattern of NK cells.

NK cells secrete a diverse range of cytokines, which alter their own properties as well as those of the cells in their microenvironment. In our previous laboratory studies, it was observed that NK-92 cells spontaneously secreted IFNγ, RANTES, and IL-10 [54], cytokines that regulate NK cell cytotoxicity. Therefore, in this study, changes in the production of these cytokines were observed.

First, the mRNA levels of the respective cytokines were assessed. It was also found that NK-92 cells naturally express the genes for IFNγ, RANTES, and IL-10. However, when exposed to the supernatant from K. pneumoniae, the relative concentration of *IL-10* mRNA was reduced. IL-10 is an anti-inflammatory cytokine [55] that reduces NK cell cytotoxicity [56,57]. The decrease in IL-10 production in this context may indicate a reduction in the autocrine and paracrine anti-inflammatory effects of NK cells. It has previously been established that IL-10 may play a role in the clearance of bacterial infections caused by *K. pneumoniae*. In an experiment involving mice with a knocked-out *IL-10* gene, increased survival in lung infections caused by this bacterium was observed in the presence of IL-10 [58]. The inhibition of IL-10 production by *K. pneumoniae* could be a mechanism for regulating the response of the host immune system.

Using CBA, we detected no change in IL-10 protein production by NK cells exposed to ESKAPE bacterial supernatants. Furthermore, these supernatants did not alter the mRNA or protein levels of IFNγ and RANTES. This contrasts with a previous study showing that a purified *S. aureus* hemolysin enhances IFNγ production [59]. This discrepancy is likely because we used a complex bacterial supernatant containing a mixture of bioactive molecules, where simultaneous stimulatory and inhibitory effects may have balanced each other out, resulting in no net change in cytokine concentration.

Notably, cytokine analysis revealed an increase in the levels of IL-10, IFNγ, and RANTES in the conditioned media obtained from NK cell cultures treated with the stress-inducing agent PMA. As the supernatants from these cultures did not exhibit similar effects, it can be inferred that they do not have a significant impact on the stress response of NK-92 cells.

Stress responses in immune system cells typically result in an increased production of pro-inflammatory cytokines [60], which may contribute to the development of protective immune responses aimed at eliminating pathogens. Therefore, bacteria may attempt to avoid inducing cellular stress in order to enhance their chances of successfully establishing an infection.

In addition, an increase in the levels of IFNγ and RANTES was observed after cell culture in the presence of TNFα. The production of TNFα and IFNγ by NK cells is closely linked: both cytokines regulate the activity of the target cell [61] and can stimulate the synthesis of each other [62]. RANTES, which is produced by NK cells, attracts increased cell migration to the site of inflammation [63]. Since TNFα is a pro-inflammatory cytokine, our findings are consistent with current understanding.

When analyzing proteins using the CBA method, no changes detected in the cytokine profile of NK cells in response to culture in the presence of supernatant were noted. This may be due to a temporal discrepancy in the synthesis of mRNA and protein. Researchers have noted a lack of convergence in the levels of mRNA and protein [64], complicating efforts to predict protein levels from mRNA data [65,66]. Cellular processes are complex. RNA degradation, translation efficiency, and post-translational regulation can lead to a disconnect between mRNA levels and the abundance of the corresponding protein. These processes can contribute to a disruption in the relationship between mRNA and protein levels. This finding provides new insights into the mechanisms underlying pathological conditions, while also opening new avenues for manipulating NK cell characteristics for therapeutic purposes.

Analysis of the NKG2A mRNA expression revealed a decrease in its relative abundance in response to exposure to *K. pneumoniae* supernatant in NK-92 cells. NKG2A is known to bind to HLA-E molecules that can be expressed on target cells, inhibiting NK cell activation [67]. There is literature evidence of positive regulation of NKG2A expression in *S. aureus* by NK cells [68].

Overall, our data demonstrate a wide range of effects of bacteria from the ESKAPE group on NK cells and their interactions with diverse target cells. The potential for regulation of NK cytotoxicity and transcriptional profile by ESKAPE bacteria was established. This finding provides new insights into the mechanisms underlying pathological conditions while also opening new avenues for manipulating NK cell characteristics for therapeutic purposes.

Ligand–receptor interactions are essential for the binding of NK cells to bacterial targets. One potential participant in this process could be the receptors of the innate immune system, such as Toll-like receptor (TLR) molecules. According to previous research, peripheral blood NK cells express TLR2 [69,70], TLR4 [69], and TLR5 [70,71]. Our goal was to assess the expression of these receptors in NK-92 cells, as each of these receptors interacts with ligands of bacterial origin to activate innate immune responses (see Table 1). By using two methods, flow cytometry and real-time polymerase chain reaction (RT-PCR), the expression levels of TLR2 and TLR5 in our samples were determined.

Traditionally, a diverse range of activating and inhibitory molecules (e.g., KIR family) are considered regulators of NK cell activity, while TLR molecules have been studied in the context of bacterial recognition without considering the effect of ligand–TLR interactions on NK cells. However, there is evidence that TLR ligand binding can alter the cytotoxic activity of NK cells [72]. Bacteria from the ESKAPE group have been shown to bind to all TLRs present on NK-92 cells and this ligand–receptor interaction may alter NK cell characteristics. In a previous study, it was demonstrated that flagellin produced by K. pneumoniae can regulate cytokine production by NK cells through binding to TLR2 and TLR5 [70]. In addition, activation of TLR2 by an agonist results in an increase in the production of IFN-γ, IL-6, and IL-8 by decidual NK cells [73], which may alter their interaction with the microenvironment during pregnancy. Therefore, the analysis of TLR expression on NK cells and their activation upon TLR binding to ligands is a priority.

As limitations of study, some points can be highlighted. In our experiments, we use cell lines and PBMC, as well as a specific strain of bacteria. While this does not entirely represent all the processes occurring in the body, it allows us to create a manageable model for studying these processes. Additionally, due to concerns regarding safety and laboratory conditions, we do not utilize live bacteria, instead employing supernatants derived from their cultivation. Furthermore, in experiments involving TLR expression, we utilized only a single time point of 24 h, which may not entirely reflect the kinetics of transcription and expression of these molecules. Our current work is of a phenomenological nature. We began with an experimental model for altering cytotoxicity. Going forward, we plan to expand our experimental panels, taking into account current limitations, and gain further insight into the impact of bacteria on NK cell characteristics.

## 4. Materials and Methods

### 4.1. Cell Cultures

For this work, we used NK-92, K-562, and JEG-3 (ATCC, Manassas, VA, USA) cells, reproducing all the main morphological, phenotypic, and functional characteristics of activated NK cells, human myelogenous leukemia, and extracellular trophoblast, respectively. All lines were cultured according to the manufacturer’s instructions (ATCC, USA). Cell viability during cultivation and in the experiment was controlled using trypan blue; it was at least 95%.

PBMC, containing NK cells, were obtained from healthy female donors in the second phase of the menstrual cycle outside the phase of exacerbation of chronic and acute infectious diseases. The study was approved by the Ethics Committee of the Federal State Institution “D. O. Ott Research Institute” (Protocol No. 124, dated 17 April 2023). PBMC were obtained by layering peripheral blood on a density gradient. After dilution with a 3:1 Hanks solution (Biolot, Saint-Petersburg, Russia), the blood was layered on a 1.077 g/mL ficoll solution (Biolot, Saint-Petersburg, Russia), then centrifuged at 350 g for 40 min. Next, the PBMC ring formed at the phase separation was selected, and the cells were washed in a Hanks solution. The PBMC were then diluted with NK cell media.

The following ESKAPE group bacterial strains were obtained from the American Type Culture Collection (ATCC, Manassas, VA, USA): Enterococcus faecium (ATCC 19434), *Staphylococcus aureus* (ATCC 29213), *Klebsiella pneumoniae* (ATCC 13883), *Acinetobacter baumannii* (ATCC 19606), *Pseudomonas aeruginosa* (ATCC 27853), and *Enterobacter* spp. (ATCC 13047). All strains were cultured on agarose medium under appropriate biosafety containment, in accordance with institutional safety protocols for handling pathogenic microorganisms.

### 4.2. Production of Conditioned Media Following the Cultivation of ESKAPE Bacteria

In order to assess the impact of ESKAPE bacterial species on NK cells, we employed conditioned media derived from their cultivation, specifically the supernatants. To generate these conditioned media, ESKAPE bacteria were cultivated at a concentration of 0.5 × 10^8^ CFU/mL in 5 mL of a complete NK cell culture medium devoid of antibiotics within a clean glass vial. The vial was then sealed with a sterile cap and incubated for 24 h at 37 °C with a CO_2_ concentration of 5%. Upon completion of the incubation period, the supernatant resulting from the bacterial cultivation was filtered through 0.22 μm pore size syringe filters (Sarstedt, Nümbrecht, Germany) to ensure purity. This process was repeated for each individual ESKAPE representative.

### 4.3. Optimal Dosage Selection for Supernatants

The day prior to the experiment, the NK-92 cell line was transplanted in accordance with the manufacturer instructions. In 24 h, the cells were transferred to a 96-well plate with round bottoms, with a concentration of 6 × 10^5^ cells per well and a volume of 50 μL of complete growth medium.

Subsequently, the supernatants were introduced into the wells, resulting in the formation of a concentration gradient. Following an additional 24 h incubation period, the cells were subjected to staining with propidium iodide solution, utilizing a working concentration of 100 ng/mL. The number of dead cells was subsequently analyzed using a FACSCanto II flow cytometer (BD, Franklin Lakes, NJ, USA). For subsequent experiments, a dilution of the supernatant following the toxic treatment was employed, ensuring that it did not compromise the viability of the NK-92 cells, as depicted in Figure 12.

### 4.4. Effect of Bacterial Supernatants on NK-92 Cell Death Stages

The cells were prepared as described above (Figure 12, Section 4.3). Then they were cultured for 24 h in a humid atmosphere at 37 °C and 5% CO_2_. After the incubation time, the cells were stained with a mixture of PI and YoPro dyes (HelloBio, Dunshaughlin, Ireland), according to the manufacturer’s instructions, after which the cell distribution was evaluated (Figure 13) using a FACSCanto II flow cytometer (BD, Franklin Lakes, NJ, USA) [74].

### 4.5. Assessment of NK-92 Cytotoxic Activity Against K-562 and JEG-3 Cells with ESKAPE Bacterial Supernatants

On the day of the experiment, cells of the K-562 or JEG-3 line were transplanted according to the standard protocol, cells of the NK-92 line were prepared as described earlier (see Figure 12, Section 4.3). A day later, the tablet with NK-92 cells was centrifuged for 10 min at 200 g, and the supernatant was removed. 100 µL of growth medium containing 6 × 10^4^ pre-stained CFSE cells of the K-562 or JEG-3 line were added to the wells, according to the manufacturer’s instructions. The contents of the wells were resuspended, centrifuged at 100 g for 10 min, after which the tablet with cells was placed in an incubator for 4 h. After the cultivation, the contents of the wells were stained with PI, after which the number of dead cells of the K-562 line was estimated using a FACSCanto II flow cytometer (BD, Franklin Lakes, NJ, USA).

### 4.6. Assessment of NK Cells in PBMC Cytotoxic Activity Against K-562 and JEG-3 Cell Lines with ESKAPE Bacterial Supernatants

On the day preceding the experiment, the cell cultures were prepared as follows: K-562 and JEG-3 cells were passaged using standard protocols. Peripheral blood mononuclear cells (PBMC), previously isolated via density gradient centrifugation (1.077 g/mL Ficoll solution, Biolot, Saint Petersburg, Russia), were plated in 96-well round-bottom plates (Sarstedt, Nümbrecht, Germany) at a density of 6 × 10^5^ cells per well in 50 µL of complete NK cell growth medium.

Supernatants were added to some of the wells in previously established concentrations, and a complete growth medium for NK cells was added to some of the wells (control wells), with the final volume in the wells being 100 µL. After one day, the target cells (K-562 or JEG-3 cells) were stained with CFSE according to the manufacturer’s instructions (Sigma-Aldrich, Saint Louis, MA, USA). At the same time, effector cells (PBMC) centrifuged 200 g for 10 min, getting rid of the medium. Then, target cells in the amount of 3 × 10^4^ or 1.5 × 10^4^ cells per 100 µL of complete medium were added to the effector cells, achieving an effector:target ratio of 10:1 or 20:1. After that, the tablet was centrifuged for 5 min at 100 g and the coculture was incubated for 4 h. After incubation, the co-culture was stained with PI according to a standard protocol and the number of dead target cells was estimated using a FACSCanto II cytometer (BD, Franklin Lakes, NJ, USA).

### 4.7. Real-Time PCR Detection of the Relative mRNA Content of Cytokine Genes in NK-92 Cells with ESKAPE Bacterial Supernatants

The cells were prepared as previously described (see Figure 12, Section 4.3), using *K. pneumoniae*, *A. baumannii*, and *Enterobacter* spp. supernatants. After 24 h, the suspension was transferred to eppendorphs and centrifuged at 200 g for 10 min to remove the filler liquid, which was analyzed using flow cytometry (see Section 4.8). Then a monophasic aqueous solution of phenol and guanidine isothiocyanate (ExtractRNA, Eurogen, Moscow, Russia) was added to the precipitate according to the manufacturer’s instructions, carefully resuspended, and further RNA was isolated using the standard phenol-chloroform method.

The RNA was then purified with isopropanol and 70% ethanol, followed by centrifugation at 12,000× *g* for 5 min at 4 °C. Further stages of sample preparation were performed on ice: after removal of the filler liquid containing ethanol, the precipitate was thoroughly dried and then dissolved in 20 µL of water without nucleases. The optical density of the RNA solution was measured using the Nanodrop One instrument (ThermoFisher, USA) to determine the concentration and purity of the RNA. Next, reverse transcription was performed to obtain complementary DNA (cDNA). 1 mL of oligo dT single–stranded deoxythymidines (Eurogen, Moscow, Russia) (100 µm) was added to a solution containing 2 micrograms of RNA, water without nucleases was brought to 9 µL and incubated for 5 min at a temperature of 70 °C. Then, 11 µL of a reaction mixture containing a buffer for the synthesis of the first cDNA strand (Eurogen, Moscow, Russia), nucleoside triphosphates (dNTP) 5 mM (Eurogen, Moscow, Russia), dithiothreitol (DTT) (Eurogen, Moscow, Russia), and mouse leukemia virus reverse transcriptase (MMLV revertase) (Eurogen, Moscow, Russia). Then, the mixture was placed in an amplifier (DNA Technologie, Moscow, Russia) to obtain complementary DNA (cDNA) in the mode of 25°—10 min, 42°—50 min, 70°—10 min, 10°—10 s. To 2 µL of cDNA obtained during reverse transcription, 23 µL of a mixture containing 5 µL of 5x qPCR mix (Eurogen, Moscow, Russia), 1 µL of reverse (R) and forward (F) (10 µm) primers, and 17 µL of H2O without nucleases were added. Then, a real-time polymerase chain reaction (PCR-RT) was started using a DT-96 detection amplifier (DNA Technology, Moscow, Russia).

Cycle 1 (1 repeat): 95°—3 min. Cycle 2 (45 repetitions): 95°—10 s, 60°—10 s, 72°—20 s.

Cycle 3 (140 repetitions): 90°—15 s.

Cycle 4 (storage)—4°.

The primers used and their sequences:

IFNγ

F: CGTGCCCACATCAAGGAGTA

R: CTTGACCTGTGGACGACTGC

RANTES

F: CGTGCCCACATCAAGGAGTA

R: CTTGACCTGTGGACGACTGC

IL-10

F: TCTTGATCATGGTCCATCCA

R: TGAACTCTGGGGTTCCATTC

The data on the expression of target genes obtained as a result of PCR were evaluated relative to the expression of the housekeeping gene GAPDH (F: GTGAACCATGAGAAGTATGACAAC

R: CATGAGTCCTTCCACGATACC).

### 4.8. Evaluation of Cytokine Production by NK-92 Cells in the Presence of Bacterial Supernatants Using Flow Cytometry

Within this study, expression of cytokines IL-10, RANTES and IFNγ were measured using commercial kits (BD, Franklin Lakes, NJ, USA) for the CBA method (Cytometric Bead Array) according to the manufacturer’s instructions.

### 4.9. Expression of NK-92 Activating and Inhibitory Receptors After Bacterial Supernatant Exposure (qPCR Analysis)

Sample preparation was carried out according to the protocol described earlier (see Section 4.7). Primers and their sequences were used:

NKG2A

F: ACTCACTCTGAGCCTTCACA

R: TCAGGGACTGTACTCTTCTGTC

NKG2C

F: CTCCAGAGAAGCTCACTGCC

R: TGTTCTGCTCCAGGAAAGGA

NKp30

F: TCTTGATCATGGTCCATCCA 

R: TGAACTCTGGGGTTCCATTC

The data on the expression of target genes obtained as a result of PCR were evaluated relative to the expression of the housekeeping gene–GAPDH (F: GTGAACCATGAGAAGTATGACAAC, R: CATGAGTCCTTCCACGATACC).

### 4.10. Flow Cytometry Analysis of Innate Immunity Receptor Expression on NK-92 Cells Exposed to Bacterial Supernatants

The cells were prepared as previously described (see Figure 1, Section 4.3). TNFα (400 U/mL), LPS (1000 ng/mL), IFNγ (500 U/100 mcl), or PMA (10 ng/100 mcl) inducers were added to some wells with cells to control cell activation. After 24 h of cultivation, the cells were treated, according to the manufacturers’ instructions, with antibodies to TLR1, TLR2, TLR4, TLR5 (BD, USA), and TLR6 (BioLegend, San Diego, CA, USA). As controls, cells treated with isotypic antibodies were used in accordance with the manufacturers’ instructions. The relative number of cells expressing the receptor or the average expression intensity (MFI) was analyzed on a FacsCantoII flow cytometer (BD, Franklin Lakes, NJ, USA).

### 4.11. Analysis of Innate Immunity Receptor Expression in NK-92 Cells Using Real-Time PCR

The day before the experiment, the cells of the NK-92 line were transplanted in accordance with the manufacturer’s instructions. After 24 h, the cells were placed in a 96-well round-bottomed plate in an amount of 6 × 10^5^ in 100 µL of full growth medium. Next, PCR was performed as described in Section 4.7 using primers and fluorescent probes:

TLR1

F–CCATCGTTGCCACCATGC

R–CCAGAAAGAATCGTGCCCACTA

Oligo–GGCTGTGACTGTGACCTCCCTCTGC + Fam + RTQ1

TLR2

F–TCTCCATCCCATGTGCGTG

R–CTGGGAGCTTTCCTGGGC

Oligo–GTTCCTGCTGATCCTGCTCACGGGG + R6G + BHQ1

TLR4

F–GCCTTCCTCTCCTGCGTG

R–GAAGGGGAGGTTGTCGGG

Oligo–TTCAGACTCCGGAGCCTCAGCCCTT + Fam + RTQ1

TLR5

F–GCTGGTGCCTTGAAGCCT

R–AGATCCTCAGGCCACCTCA

Oligo–ATGGTGGTGGTTGGGTCCTTGTCCC + R6G + BHQ1

TLR6

F–ACCTACATCCCCTCGGCA

R–GGTGCACAGTGTCCCCTC

Oligo–GGCCTCCCTGCATCCCAGTGGAAAG + Fam + RTQ

GapdH

F–AAGGGCATCCTGGGCTAC

R–GTGGAGGAGTGGGTGTCG

Oligo–TGAGCACCAGGTGGTCTCCTCTGAC + ROX + BHQ2

## 5. Conclusions

Within this study, we examined the interaction between NK cells and bacteria of the ESKAPE group. The influence of bacterial supernatants on functional, phenotypical, and secretory characteristics of NK cells was assessed.

The interaction of bacteria with NK cells is likely mediated through ligand–receptor binding. Given that the expression of TLR2 and TLR5 on NK-92 cells were confirmed with two methods, these cells are capable of responding to bacterial signals.

Furthermore, this results in a multidirectional alteration in the cytotoxicity of NK cells towards various targets: supernatants from *K. pneumoniae*, *Enterobacter species*, or *S. aureus* reduced the cytotoxic capacity of NK-92 cells against K-562, whereas pretreatment of effector cells with supernatants from *E. faecium*, *A. baumannii*, or *Enterobacter species* enhanced the cytotoxic activity of NK-92 against JEG-3. The findings are also applicable to NK cells in PBMC fractions. This may be reflective of the pathogenesis of certain processes, which are based on disrupted interactions between NK cells and the microenvironment. In order to elucidate the mechanisms underlying this phenomenon, we evaluated the effect of supernatants on cytokine production by NK cells. Despite not demonstrating any changes in protein synthesis, our results show a reduction in amount of IL-10 mRNA levels in NK-92 cells when exposed to *K. pneumoniae* supernatants. Additionally, a decrease in NKG2A mRNA levels under the same conditions was noticed.

Our findings indicate that NK cells play a role in antibacterial immune responses. Bacteria actively influence the functions of NK cells, which in turn can be affected by bacterial products.

## Figures and Tables

**Figure 1 ijms-26-08449-f001:**
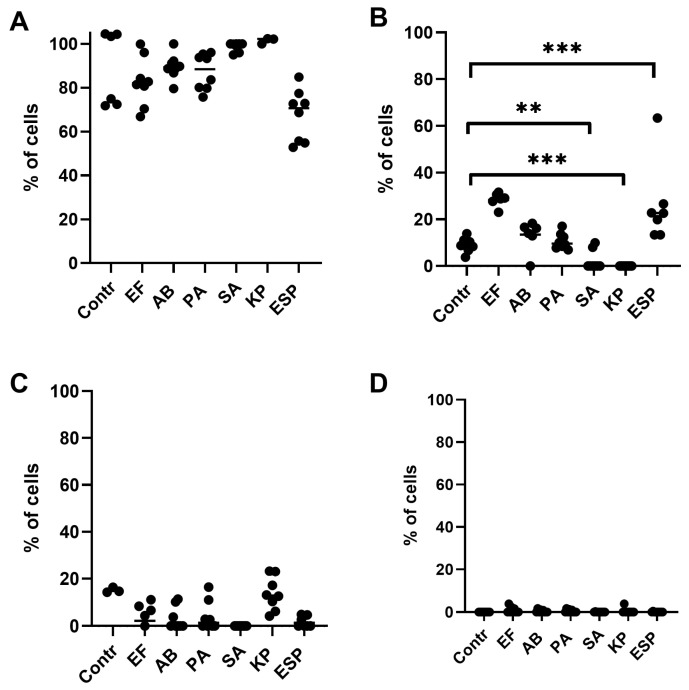
Relative number of NK-92 cells that underwent apoptosis and necrosis during cultivation for 24 h in the presence of supernatants of the ESKAPE group bacteria. (**A**)—alive NK-92. (**B**)—NK-92 cells in a state of early apoptosis. (**C**)—NK-92 cells in a state of late apoptosis. (**D**)–NK-92 cells in a state of necrosis. Contr—intact cells of the NK-92 line, cells of the NK-92 line, pre-cultivated with supernatants of bacteria of the ESKAPE group (*P. aeruginosa (PA)*, *E. faecium (EF)*, *K. pneumoniae (KP)*, *A. baumanii (AB)*, *Enterobacter* spp. *(Esp)*, *S. aureus (SA)*). The number of assays = 4 with two technical replicates in each. Stars indicate statistical significance: ** -*p* < 0.01, *** *-p* < 0.001. Bars show the median.

**Figure 2 ijms-26-08449-f002:**
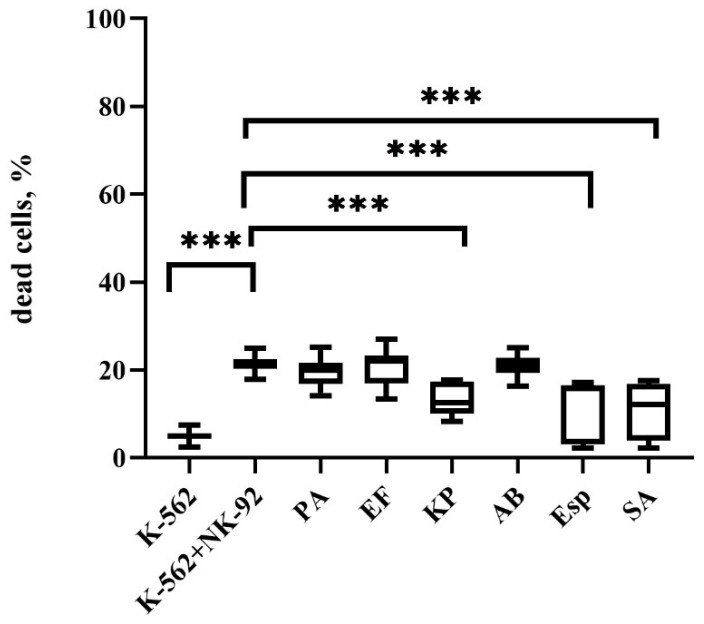
Cytotoxic activity of NK-92 cells against K-562 targets after exposure to bacterial supernatants. K-562: Baseline cell death; K-562+NK-92: Cell death after co-culture with untreated NK-92 cells; Cell death after NK-92 cells were pre-incubated with supernatants from *P. aeruginosa (PA)*, *E. faecium (EF)*, *K. pneumoniae (KP)*, *A. baumannii (AB)*, *Enterobacter* spp. *(Esp)*, and *S. aureus (SA)*. The number of assays = 5 with two technical replicates in each. Stars indicate statistical significance: *** *-p* < 0.001. Bars in boxes show the median.

**Figure 3 ijms-26-08449-f003:**
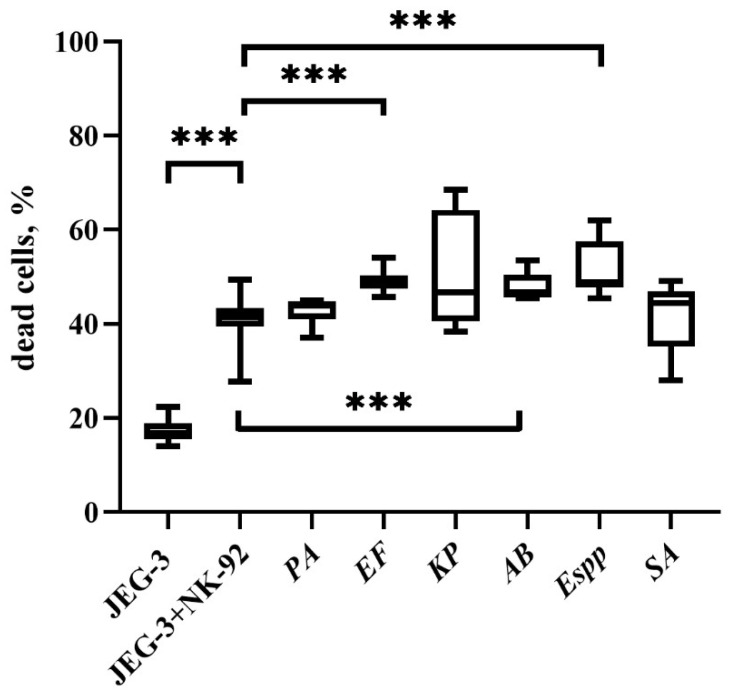
NK-92 cell cytotoxicity against JEG-3 targets following bacterial supernatant exposure. JEG-3: Baseline cell death, JEG-3+NK-92: Cell death after co-culture with untreated NK-92 cells. Cell death after NK-92 cells were pre-incubated with supernatants from *P. aeruginosa (PA)*, *E. faecium (EF)*, *K. pneumoniae (KP)*, *A. baumannii (AB)*, *Enterobacter* spp. *(Espp)*, and *S. aureus (SA).* The number of assays = 5 with two technical replicates in each. Stars indicate statistical significance: *** *-p* < 0.001. Bars in boxes show the median.

**Figure 4 ijms-26-08449-f004:**
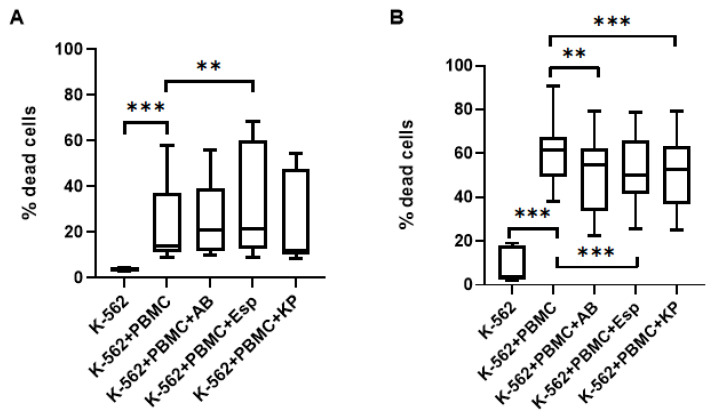
PBMC-mediated cytotoxicity against K-562 cells following exposure to ESKAPE bacterial supernatants at effector:target ratios of (**A**) 10:1 and (**B**) 20:1. K-562: Baseline cell death; K-562+PBMC: Cytotoxic activity of untreated PBMC against targets; PBMC cytotoxicity after pre-incubation with supernatants from ESKAPE pathogens: *K. pneumoniae (KP)*, *A. baumannii (AB)*, and *Enterobacter* spp. *(Esp).* The number of assays = 6 with two technical replicates in each. Stars indicate statistical significance: ** -*p* < 0.01, *** -*p* < 0.001. Bars in boxes show the median.

**Figure 5 ijms-26-08449-f005:**
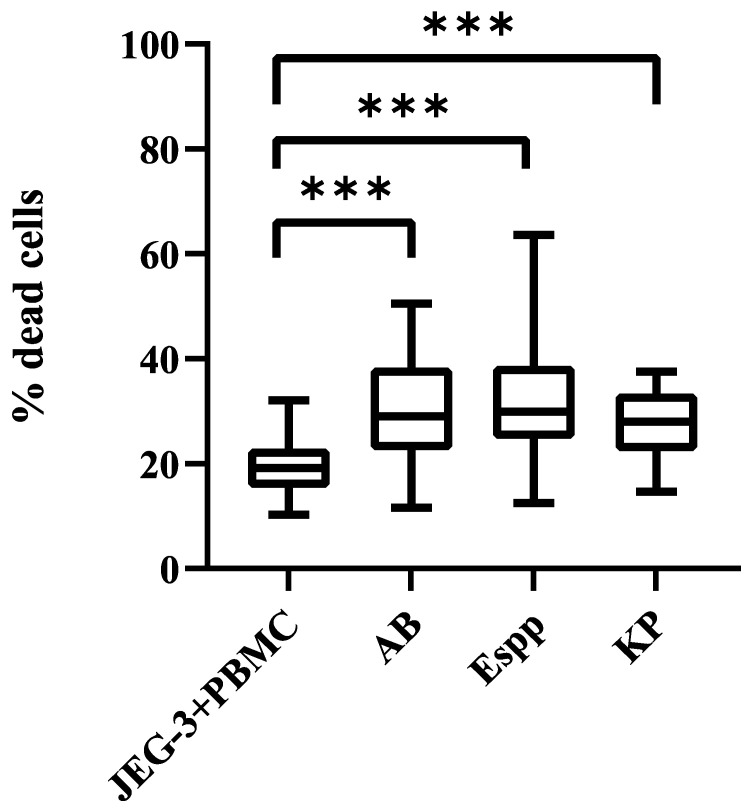
PBMC-mediated cytotoxicity against JEG-3 cells following exposure to ESKAPE bacterial supernatants. JEG-3+PBMC: Cytotoxic activity of untreated PBMC against targets; PBMC cytotoxicity after pre-incubation with supernatants from ESKAPE pathogens: *K. pneumoniae (KP)*, *A. baumannii (AB)*, and *Enterobacter* spp. *(Espp).* The number of assays = 6 with two technical replicates in each. Stars indicate statistical significance: *** -*p* < 0.001. Bars in boxes show the median.

**Figure 6 ijms-26-08449-f006:**
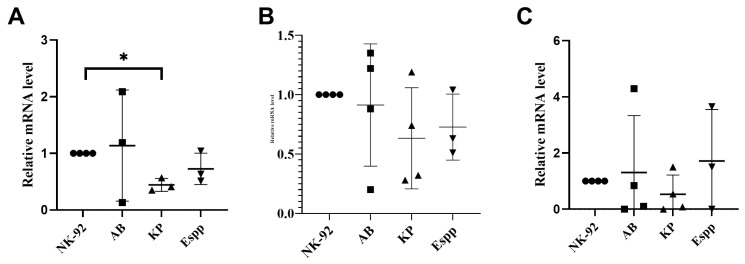
Effects of bacterial supernatants on cytokine gene expression in NK-92 cells. Relative mRNA levels of (**A**) *IL-10,* (**B**) *IFNγ*, and (**C**) *RANTES* for NK-92-untreated NK-92 cells, as well as *A. baumannii* (AB), *Enterobacter* spp. (Espp), and *K. pneumoniae* (KP). The number of assays = 4 with graphs showing the average of two technical replicates. Stars indicate statistical significance: * -*p* < 0.05. Bars show the median.

**Figure 7 ijms-26-08449-f007:**
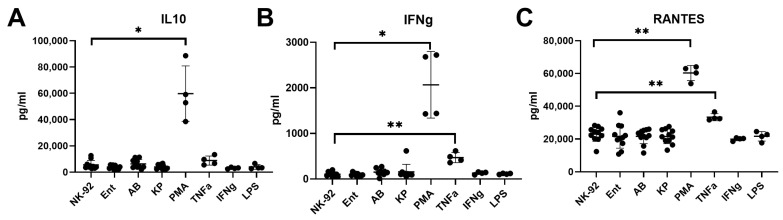
Effect of bacterial supernatants on the content of cytokines IL-10 (**A**), IFNγ (**B**), and RANTES (**C**) in conditioned media after cultivation of NK-92 cells under various conditions. Designation in the figure: relative mRNA content in intact cells (NK-92); relative content in cells of the NK-92 line previously incubated with the supernatants *A. baumannii (AB)*, *Enterobacter* spp. *(Espp)*, *K. pneumoniae (KP)*, PMA, IFNγ, and LPS. The number of assays conducted with supernatants = 5 with two technical replicates in each. The number of assays conducted with other inductors = 4 with graphs showing the average of two technical replicates. Stars indicate statistical significance: * -*p* < 0.05, ** -*p* < 0.01. Bars show the median.

**Figure 8 ijms-26-08449-f008:**
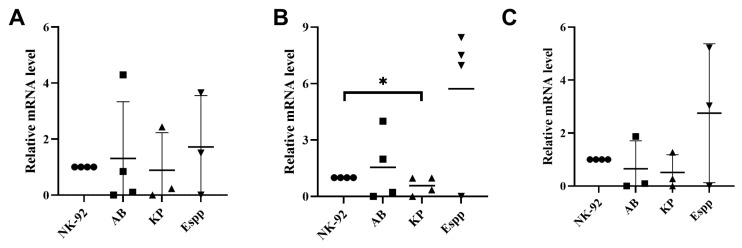
NK cell receptor gene expression modulation by bacterial supernatants. Relative mRNA levels of (**A**) NKG2C, (**B**) NKG2A, and (**C**) NKp30 for NK-92-untreated NK-92 cells, as well as *A. baumannii* (AB) *Enterobacter* spp. (Espp) *K. pneumoniae* (KP). The number of assays = 4 with graphs showing the 2^−ddCT^. Stars indicate statistical significance: * -*p* < 0.05. Bars show the median.

**Figure 9 ijms-26-08449-f009:**
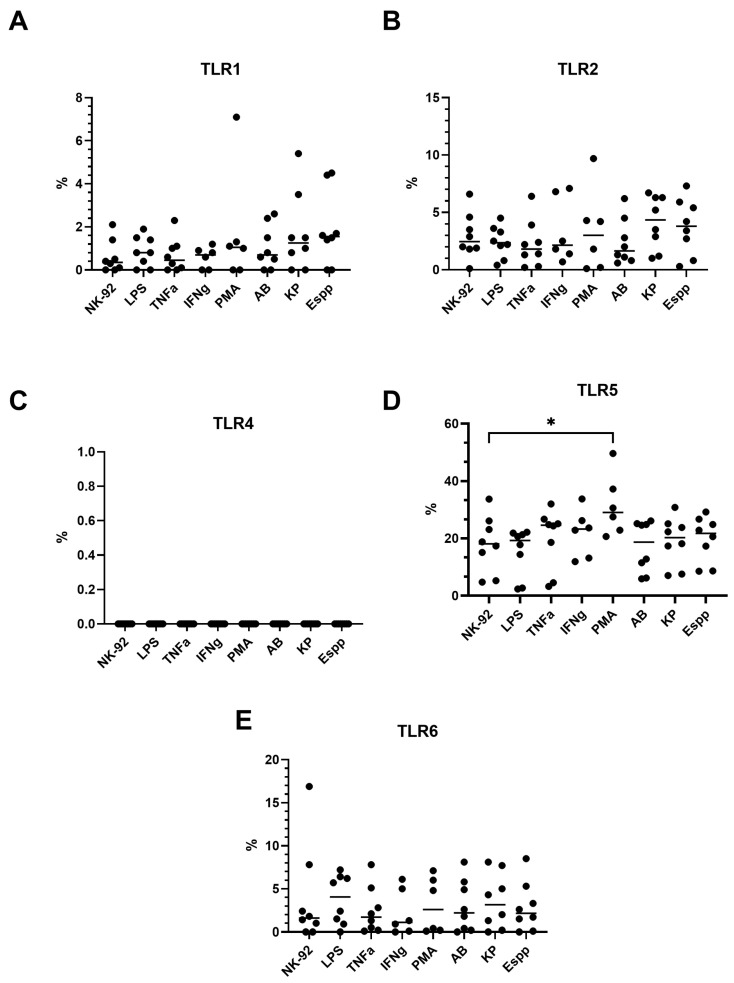
TLR expression profiles in NK-92 cells following bacterial supernatant incubation. (**A**) TLR1, (**B**) TLR2, (**C**) TLR4, (**D**) TLR5, (**E**) TLR6 expression levels in NK-92-untreated NK-92 cells; NK-92 cells exposed to supernatants from *A. baumannii (AB)*, *Enterobacter* spp. *(Espp)*, or *K. pneumoniae (KP)*; or stimulated with LPS, TNFα, IFNγ, or PMA. The number of assays = 4 with two technical replicates in each. The number of assays conducted with PMA and IFN = 3 with two technical replicates in each. Stars indicate statistical significance: * -*p* < 0.05. Bars show the median.

**Figure 10 ijms-26-08449-f010:**
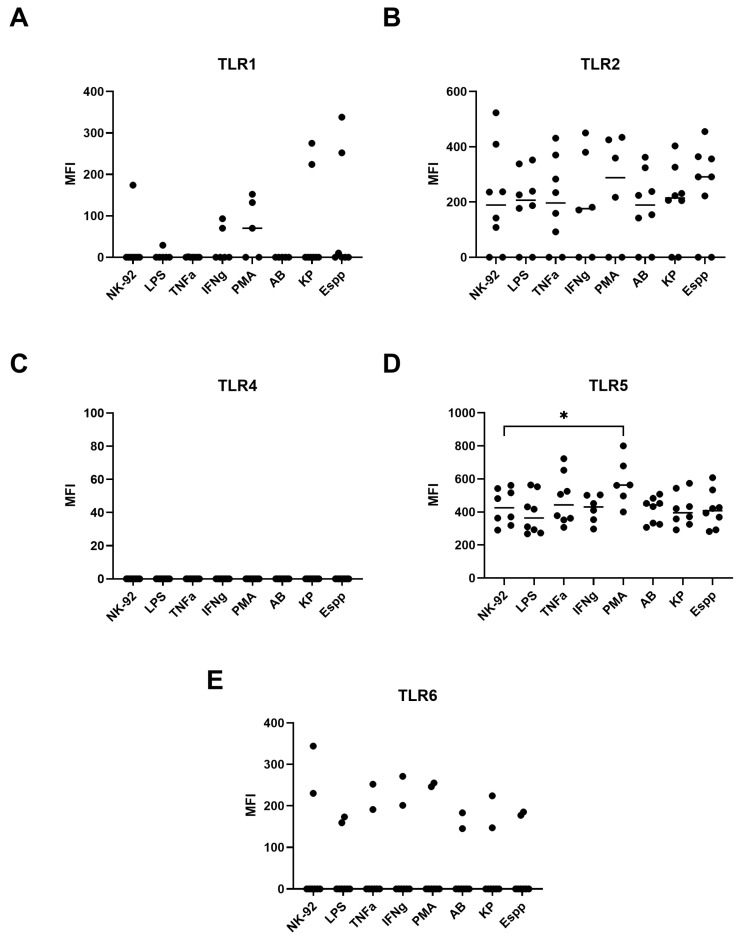
Expression levels of (**A**) TLR1, (**B**) TLR2, (**C**) TLR4, (**D**) TLR5, and (**E**) TLR6 in NK-92 cells. Data show relative mRNA content in untreated NK-92 cells (NK-92) versus cells treated with supernatants from *A. baumannii (AB)*, *Enterobacter* spp. *(Espp)*, and *K. pneumoniae (KP)*, or stimulated with LPS, TNFα, IFNγ, or PMA. The number of assays = 4 with two technical replicates in each. The number of assays conducted with PMA and IFN= 3 with two technical replicates in each. Stars indicate statistical significance: * -*p* < 0.05. Bars show the median.

**Figure 11 ijms-26-08449-f011:**
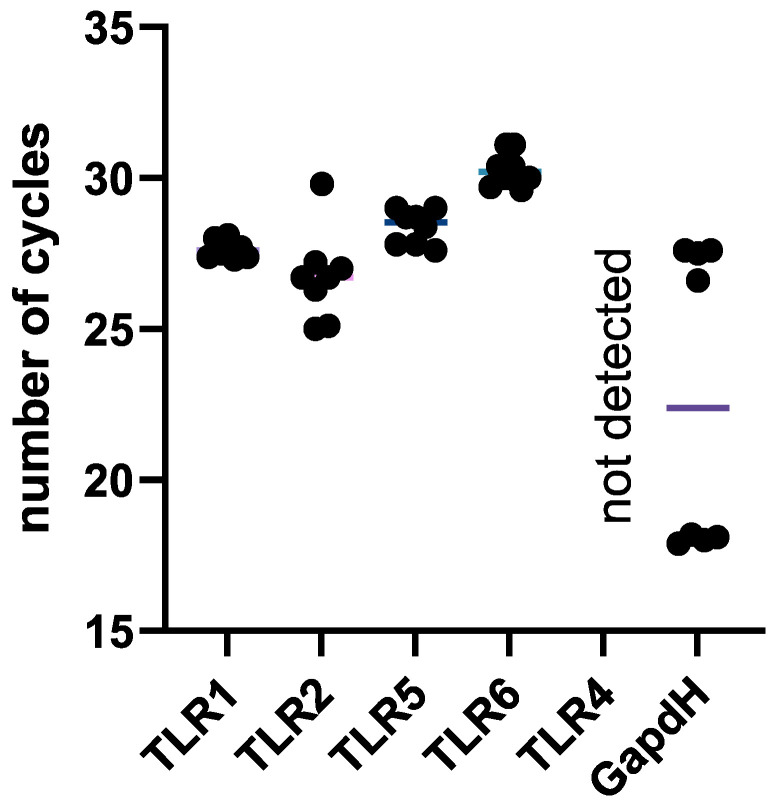
Cycle threshold (Ct) values for TLR1, TLR2, TLR4, TLR5, TLR6, and GAPDH detection by qPCR. The number of assays = 4 with two technical replicates in each.

**Figure 12 ijms-26-08449-f012:**
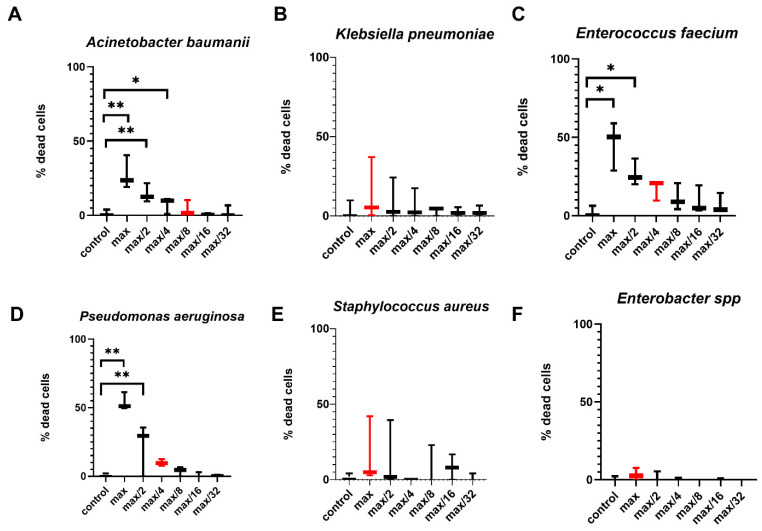
Selection of the optimal concentration of supernatants for further experiments. (**A**)—the effect of the *A. baumanii* supernatant on the viability of NK-92 cells. (**B**)—the effect of *K. pneumoniae* supernatant on the viability of NK-92 cells. (**C**)—the effect of the *E. faecium* supernatant on the viability of NK-92 cells. (**D**)—the effect of the *P. aeruginosa* supernatant on the viability of NK-92 cells. (**E**) —the effect of the *S. aureus* supernatant on the viability of NK-92 cells. (**F**)—the effect of *Enterobacter* spp. supernatant on the viability of NK-92 cells. The red color indicates the concentration of the supernatants selected for further experiments, which was used for all further experiments. The number of assays = 3 with two technical replicates in each. Stars indicate statistical significance: * -*p* < 0.05, ** -*p* < 0.01. Bars in boxes show the median.

**Figure 13 ijms-26-08449-f013:**
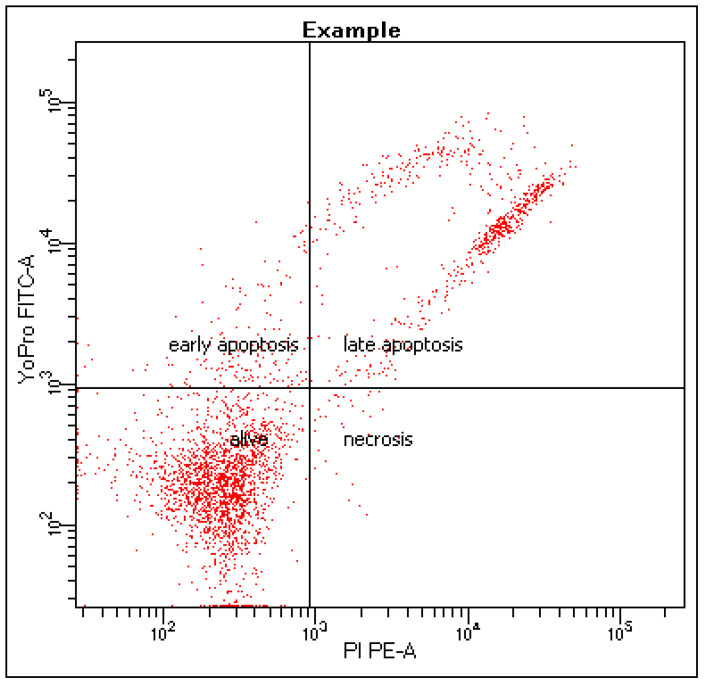
Distribution of NK-92 cells: viable cells (alive), cells in a state of early apoptosis or late apoptosis and necrosis.

**Table 1 ijms-26-08449-t001:** Ligands of TLR molecules found on NK-92 cells.

**Receptor**	**Ligand**	**Potential Object of Interaction**
TLR2	Peptidoglycan, triacillipopeptide	Gram-positive, Gram-negative bacteria
TLR5	Flagellin	Gram-positive, Gram-negative bacteria

## Data Availability

The data presented in this study are available on request from the corresponding author.

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
