# Peer review of "Modulation of NK Cell Properties by ESKAPE Group Bacteria"

_ijms, 2025, doi:10.3390/ijms26178449_

Round 1
Reviewer 1 Report
Comments and Suggestions for Authors
This study titled “Modulation of NK cell properties by ESKAPE group bacteria” focuses on effects of bacterial supernatants on NK cells characteristics after coculturing. It concludes that bacteria can modify NK cells features and their interaction with cells in the microenvironment.
The study has some novel findings, and its writing was very good. Therefore, I would recommend its publication in this journal after a major revision to correct the following:
- The findings of the study are accurately summarized in the Abstract. Minimizing the use of “we” would be good.
- Keywords should be revised and arranged in alphabetical order for greater clarity and consistency.
- Line 31, “six prokaryotic organisms”, it would be more accurate to correct it to “six bacterial species”.
- Line 34, “responsible for a significant number of nosocomial infections [1].”, writing a few examples of these infections could be beneficial for readers.
- Line 65, “Enterobacter spp и Klebsiella spp”, should be corrected to “Enterobacter and Klebsiella spp.” Similarly, in line 68, spp should not be italicize and a period should be added “spp.”
- Lines 71 and 72, “women with a history of recurrent miscarriages tend to have an endometrial microbiota dominated by Acinetobacter species [15].”, what about other bacterial species belonging to ESKAPE group bacteria? Do they contribute to such incidents? Please expand this section to cover evidences on the above point from literature.
- Line 76, “prokaryotic organisms”, please see my comment (3) above.
- Line 82, “we”, please see my comment (1) above. Please check the whole manuscript to modify the writing as necessary.
- Line 83, “sawno”, please correct this.
- Lines 84, 84, and in other lines in the manuscript, bacterial species should be italicized. Please check the whole manuscript to correct the scientific writing.
- 1, legend should be informative. In B, what do the stars, and bars, represent? All legends (of all figures) should be checked and corrected.
- Lines 125 – 127, “In addition, the content of NK cells in the PBMC fraction is about 10%, so we also decided to increase the amount of PBMC in the experiments to increase the effector ratio (PBMC): the target is up to 20:1.”, setting up this ratio need additional explanations for clarity.
- 11, “samples with Ct <35 cycles were considered positive.”, this needs justification.
- Lines 229 and 230, “The results obtained from YoPro staining confirm the selection of supernatant concentrations for further experiments.”, here, selecting criteria should be elucidated.
- Lines 286 – 288, “It is possible that these bacteria of ESKAPE group disrupt the balance required for optimal interaction between trophoblast and NK cells, leading to impaired fertility.”, is there any evidence from published papers on this possibility?
- The limitations of the study should be highlighted in the appropriate sections.
- In the materials and methods sections, ATCC numbers should be provided.
- Proofreading the entire manuscript is required to correct grammatical errors and improve sentence structure for clarity and flow.
Author Response
Dear Reviewer,
Thank you for taking the time to review our work. We appreciate your attention to detail and have carefully considered all of your comments.
The findings of the study are accurately summarized in the Abstract. Minimizing the use of “we” would be good.
Line 82, “we”, please see my comment (1) above. Please check the whole manuscript to modify the writing as necessary.
We thank the reviewer for this suggestion. We agree that varying sentence structure can enhance readability. The use of active voice with "we" is often recommended in modern scientific writing (e.g., by Nature, Science, and the ACS Style Guide) for its clarity and directness, as it unambiguously states who performed the action and avoids overly complex passive constructions. However, we acknowledge that styles and preferences can vary. In accordance with the reviewer's suggestion, we have carefully revised the entire manuscript to significantly reduce the use of "we" where possible.
Keywords should be revised and arranged in alphabetical order for greater clarity and consistency.
Changes to keywords list were made, see line 22.
Line 31, “six prokaryotic organisms”, it would be more accurate to correct it to “six bacterial species”.
Line 76, “prokaryotic organisms”, please see my comment (3) above.
Thank you, we fixed this, see lines 30, 76.
Line 34, “responsible for a significant number of nosocomial infections [1].”, writing a few examples of these infections could be beneficial for readers.
Thank you, we added this information to the text, see lines 33-35.
Lines 71 and 72, “women with a history of recurrent miscarriages tend to have an endometrial microbiota dominated by Acinetobacter species [15].”, what about other bacterial species belonging to ESKAPE group bacteria? Do they contribute to such incidents? Please expand this section to cover evidences on the above point from literature.
Thank you, we added information to the text, see lines 72-75.
Line 83, “sawno”, please correct this.
Thank you, we fixed the typo.
Lines 84, 84, and in other lines in the manuscript, bacterial species should be italicized. Please check the whole manuscript to correct the scientific writing.
Thank you, we fixed the species formatting throughout the text.
legend should be informative. In B, what do the stars, and bars, represent? All legends (of all figures) should be checked and corrected.
All legends have been changed and expanded with additional information.
Lines 125 – 127, “In addition, the content of NK cells in the PBMC fraction is about 10%, so we also decided to increase the amount of PBMC in the experiments to increase the effector ratio (PBMC): the target is up to 20:1.”, setting up this ratio need additional explanations for clarity.
The researchers described the use of a 1:5 or 1:10 effector:target ratio (10.3389/fimmu.2020.01851, 10.1016/j.humimm.2011.08.006) to evaluate the interaction between PBMC and targets. However, it is also possible to increase the number of effectors up to a 20:1 ratio (10.3389/fimmu.2020.01851 10.3791/56191), as this allows for a higher content of NK cells in the experiment. In future, we intend to repeat these studies with NK cells that have been purified from PBMC.
“samples with Ct <35 cycles were considered positive.”, this needs justification.
The amplification results are considered positive when detecting the product before the 40-45 cycle PCR Cycling Parameters—Six Key Considerations for Success | Thermo Fisher Scientific - RU, PCR cycle steps. The content was recognized as positive upon detection of the PCR product earlier that cycle 35, whereas in negative control samples the signal was absent before the 45th cycle. With the phrase "the content was identified as positive prior to cycle 35 of the PCR process", we wished to emphasize that the PCR products had reached a positive result prior to the specified cycle. At this time, this phrase has been removed to avoid confusion.
Lines 229 and 230, “The results obtained from YoPro staining confirm the selection of supernatant concentrations for further experiments.”, here, selecting criteria should be elucidated.
Previously, we had based our decision on the results from the PI assay alone – that is, we selected the dose of supernatant that did not result in an increase in the number of PI-positive cells compared to the control group. Next, we decided to verify the selected dose using the YoPro/PI combination. In this assay, we expected to observe no increase in the number of viable cells or cells undergoing late apoptosis or necrosis relative to the control, as this is the function of the PI stain. Since this was the case, we believe that it indicates the appropriate choice of dose for further experimentation.
Lines 286 – 288, “It is possible that these bacteria of ESKAPE group disrupt the balance required for optimal interaction between trophoblast and NK cells, leading to impaired fertility.”, is there any evidence from published papers on this possibility?
There is no such evidence. According to the facts described, bacteria are often found in various unfavorable reproductive processes. Additionally, our laboratory has been actively investigating the effects of various factors on the interactions between NK cells and trophoblasts. Research has shown that a disruption of this process, particularly a shift in the cytotoxic balance between NK cells and trophoblasts, can play a significant role in miscarriage. Consequently, based on literature and our own findings, we propose this hypothesis.
The limitations of the study should be highlighted in the appropriate sections.
We added limitations of study as a part of discussion, line 410-420.
In the materials and methods sections, ATCC numbers should be provided.
Done.
Proofreading the entire manuscript is required to correct grammatical errors and improve sentence structure for clarity and flow.
Done.
Thank you for your continued interest in our research and for your assistance in improving the manuscript.
Reviewer 2 Report
Comments and Suggestions for Authors
The article refers to the effect of resistant bacteria extracellular products and the effects on NK-92 cell line and PBMC and cytotxic response against K562 and JEG3 cell lines. The effects are evident against JEG3 cells than K562 cells and the stimulation of the cells was analyzed by the production of cytokines. In general, the rationale is appropiate and the methodology is standard. There are however, few issues that may enhance the manuscript. One of them referes to the kinetic of the priming of NK92 and PBMC cells prior the cytotoxic response. The time of priming may be crucial. In addition, the kinetic on Toll receptor transcription and expression would be important to analyze further as well the possible inhibition of the process. The discussion should provide a plan for future perspectives based on the mechanism proposed.
Minor comment please add the number of assays in the legends of the figures.
Author Response
Dear reviewer!
Thank you for taking the time to review our work. We appreciate your attention to detail and have carefully considered your comments. We would like to specifically address one of them:
There are however, few issues that may enhance the manuscript. One of them refers to the kinetic of the priming of NK92 and PBMC cells prior the cytotoxic response. The time of priming may be crucial. In addition, the kinetic on Toll receptor transcription and expression would be important to analyze further as well the possible inhibition of the process. The discussion should provide a plan for future perspectives based on the mechanism proposed
We have taken note of your feedback regarding the kinetics of TLR transcription and expression. The aim of this study was to detect phenomenon of changing NK cells properties by bacteria. The aim of the TLR assessment experiments as a part of study was to elucidate the possible mechanisms by which bacteria influence NK cell characteristics. This phenomenon has not been adequately explored in the literature and the expression of TLR NK-92 has not been previously reported. In the future, we intend to evaluate both mRNA and protein levels at different time points during NK-92 interactions with bacteria or their culture supernatants in order to gain a more comprehensive understanding of the process. We added some explanations as a part of discussion, lines 410-420.
We have also included information regarding the number of experiments in the captions for the figures.
Thank you for your continued interest in our research and for your assistance in improving the manuscript.
Round 2
Reviewer 1 Report
Comments and Suggestions for Authors
I have checked the revised version of the manuscript and found that all requested corrections were made in the new version. Therefore, I would recommend its publication in the journal.